# Most Common Long COVID Physical Symptoms in Working Age Adults Who Experienced Mild COVID-19 Infection: A Scoping Review

**DOI:** 10.3390/healthcare10122577

**Published:** 2022-12-19

**Authors:** Zoe Mass Kokolevich, Melissa Crowe, Diana Mendez, Erik Biros, Jacqueline Elise Reznik

**Affiliations:** 1Cohort Doctoral Studies Program, Australian Institute of Tropical Health and Medicine, James Cook University, Townsville, QLD 4811, Australia; 2Cohort Doctoral Studies Program, Australian Institute of Tropical Health and Medicine & Subject Coordinator, James Cook University, Townsville, QLD 4811, Australia; 3College of Medicine and Dentistry, James Cook University, Townsville, QLD 4811, Australia

**Keywords:** long COVID, physical symptoms, mild COVID-19, adults

## Abstract

Background: One-third of patients who recover from COVID-19 present with long COVID. Their symptoms are broad, affecting their physical functioning and, ultimately, their quality of life. Many of those individuals who develop long COVID, possibly from a mild COVID-19 infection, are in the 18–65 age group. This prolongation of malaise directly influences national workforce economies. Objectives: To summarise the commonly reported physical symptoms of long COVID in order to inform potential adjustments in healthcare for the employable population. Methods: The Embase, CINAHL, Medline, SCOPUS, and WHO COVID-19 databases were searched. The study selection process was based on the PRISMA guidelines. The extracted data were synthesised and presented narratively. Results: 7403 studies were accessed, comprising 60 cohort studies and 10 case series/studies, representing 289,213 patients who met our criteria. The most frequently reported physical symptoms were fatigue (92%), shortness of breath (SOB) (81.8%), muscle pain (43.6%), and joint pain (34.5%). Conclusions: The range of reported physical symptoms was broad and varied; the main ones being fatigue, breathlessness/SOB, and pain. Similarities observed between long COVID and other post-acute infection syndromes may help formulate protocols to manage and promote recovery for long COVID patients. Inconsistencies were evident, particularly with a lack of adherence to the standardised definitions of long COVID.

## 1. Introduction

The Coronavirus Pandemic (COVID-19) was caused by the emergence of the severe acute respiratory syndrome coronavirus (SARS-CoV-2) in humans and continues to spread, globally, in a number of variant forms. To date, there have been in excess of 550 million confirmed cases and over six million deaths in more than 200 countries [1]. The COVID-19 pandemic dramatically affected the world’s economy [2,3] and impacted all aspects of the socioeconomic spectrum. The global response to the pandemic, in terms of government policy, vaccination development and availability, slowed the spread of COVID-19 and its effects [4,5,6]. The initial priorities of health services around the world were to minimise the number of deaths and reduce the burden on public health systems. However, by August 2020, the focus had shifted to the long-term health effects for the survivors [7]. Published reports indicated that approximately 10–30% of COVID-19 patients presented with persistent complaints for at least two months post-onset of the acute infection [8,9]. These persistent complaints or newly presenting symptoms were referred to as long COVID when they could not be attributed to any other disease [10,11]. 

Defining the long-term clinical presentation of the COVID-19 infection has been challenging. There are overlaps between the types of symptoms, their timing, and their fluctuating nature. A definition of the clinical case presentations of this novel condition was developed in the National Institute for Health and Care Excellence (NICE) Guidelines (2020) to allow practitioners to identify and diagnose the long-term effects of COVID-19 (long COVID). In October 2021, the WHO developed their clinical case definition of the post-COVID-19 condition using a Delphi consensus [9]. Figure 1 is a visual classification of the definition of long COVID based on the timing from symptom onset, according to the NICE guidelines and the WHO.

As shown in Figure 1, the NICE guidelines (2020) state that ‘long COVID’ is commonly used to describe signs and symptoms that continue or develop after acute COVID-19. It includes both ongoing symptomatic COVID-19 (from 4–12 weeks) and post-COVID-19 syndrome (12 weeks or more) [11]. The WHO’s definition states, “Post COVID-19 condition occurs in individuals with a history of probable or confirmed SARS-CoV-2 infection, usually 3 months from the onset of COVID-19, with symptoms lasting for at least 2 months that cannot be explained by an alternative diagnosis” [9].

The frequency of reported persistent physical symptoms varies between 10% and 35% of all COVID-19 survivors [12]. Early research studies primarily focused on patients who had been hospitalised during the acute phase of their infection [13], but more recent studies have shown that long COVID symptoms can be independent to the severity of the initial acute infection [14]. More than 300 symptoms of long COVID have been identified and documented [15]. These symptoms are varied (Table 1) and may affect a broad range of body systems, in some instances, leading to impairment and limitations of “body structures” and “functions”, as defined by the International Classification of Functioning, Disability and Health framework (ICF) [16]. In turn, these impairments can affect the individual’s quality of life (QoL) and day-to-day activities, including physical functioning and subsequent participation in the workforce [17,18].

A considerable proportion of people with long COVID become dependent on others for personal activities of daily living and often carers are family members who may otherwise be in the workforce [19,20]. Recent literature has highlighted the disparity of the impact of long COVID on different socioeconomic groups [21]. This may have a particularly negative effect in countries such as the United States of America (USA), where health insurance is linked to a person’s employment and the loss of a job leads to the total loss of the family’s health insurance [22].

More than 80% of those who contracted COVID-19 were aged between 18 and 65 years but as they experienced milder symptoms, they were not hospitalised during the acute phase of their infection [23]. This age group corresponds to the “working age” population in many countries, e.g., the United Kingdom (UK), where it accounts for almost two-thirds of the total populace [24].

The rapid sharing of academic and scientific information concerning COVID-19 and the subsequent condition of long COVID has been an effective way to reduce public alarm and provide real time guidance for clinicians managing patients. It has also enabled policy makers to propose future developments and evaluate the possible effectiveness of various interventions [25]. Over 6000 papers have been published on long COVID since the beginning of the pandemic; however, there are still many unanswered questions regarding its prevalence, frequency, predisposing risk factors, treatment, and rehabilitation [8]. This scoping review aims to document the knowledge gained, to date, from the peer-reviewed published literature in relation to the frequency of the reported physical symptoms of long COVID and to identify any important gaps in the literature. The term long COVID will be used in this paper and will be inclusive of the term post-COVID-19 and other synonyms, as detailed in the search strategy section.

## 2. Materials and Methods

This scoping review was conducted on 5 January 2022, following the five-step framework outlined by Arksey and O’Malley [26], and aimed to answer the following question, formulated using the PICo framework [27].

“What are the commonly reported physical symptoms of long COVID in adults aged between the ages 18–65, who were not hospitalised during their acute COVID-19 infection?”

### 2.1. Identifying Relevant Studies

The PRISMA (Preferred Reporting Items for Systematic Reviews and Meta-Analyses) strategy was followed [28] to identify all relevant studies. (Figure 2—PRISMA Flow Chart).

### 2.2. Search Strategy

The search strategy employed was original and was developed with advice from a senior librarian. A comprehensive search of the most relevant literature was undertaken using the following databases: Embase (via Ovid), CINAHL, Medline (via Ovid), SCOPUS, and WHO COVID-19 Database. No filters were applied regarding the starting dates due to the novel topic and the search was terminated on 3 January 2022. The search strategy was initially created in Medline and then translated into the other databases. Keywords were used in Medline as no MESH terms for long COVID were available at the time of searching. The key words of the identified studies were examined to make sure no alternatives were missing. “Citation chaining” was also utilised to ensure all relevant papers were included. 

The search terms and Boolean operators were applied as follows:

“long COVID” OR “Long-COVID” OR “Post-COVID” OR “Long COVID-19” OR “Long coronavirus disease 19” OR “Long coronavirus disease 2019” OR “Long coronavirus disease-19” OR “Long Sars Cov 2” OR “Long Sars coronavirus 2” OR “Long sars-cov-2” OR “Post 2019 novel coronavirus” OR “Post 2019 ncov” OR “Post COVID-19” OR “Post covid-19 *” OR “Post covid19” OR “Post coronavirus disease 19” OR “Post coronavirus disease 2019” OR “Post coronavirus disease-19” OR “Post Sars cov 2” OR “Post Sars coronavirus 2” OR “Post sars-cov-2” OR “Chronic COVID-19” OR “post-COVID-19 syndrome” OR “post-acute sequelae of COVID-19” OR “PASC” OR “chronic COVID syndrome” “CCS” OR “post-COVID-19” OR “protracted Covid-19” OR “Covid long haulers” OR “long haul COVID-19”.

AND

“sign*” OR “symptom*” OR “Clinical presentation”.

A backward citation check was undertaken by searching the included studies’ bibliographies, and forward citation checking was achieved using SCOPUS.

### 2.3. Study Selection

This scoping review focused on individuals with long COVID who did not require hospitalisation during their acute COVID-19 infection and/or who otherwise had mild COVID-19 symptoms. To ensure the inclusion of all physical symptoms, “non-hospitalised” was *not* added as an original search term, as pilot searches had shown many relevant studies were lost when this search term was included. Only peer-reviewed, primary research studies in English were included (Table 2). However, due to the novelty of the COVID-19 pandemic, there were no date restrictions. Although some studies did include both hospitalised and non-hospitalised patients, in cases where more than 25% of the participants were hospitalised, these studies were not included. After completing all of the database searches, the citations were collected and entered into EndNote X9 bibliographic management software, where duplicate citations were removed using the SR Accelerator [24]. The first stage screening included titles and abstracts using the SR Accelerator Screenatron function [29]. Two reviewers (ZMK and JER) carried out the screening independently. Disagreements were discussed, and a consensus was reached between the reviewers. Complete text copies were obtained and independently screened by two reviewers (JER and ZMK). All studies included in the review were selected after a consensus was reached between both reviewers. Following the retrieval of these studies, manual searching was undertaken to ensure comprehensiveness.

### 2.4. Charting the Data

Data extraction and validation were carried out manually by transferring the relevant information from each study into a separate Microsoft Excel (Microsoft Office Professional Plus 2010, Version 14.0.7128.5000) spreadsheet. The relevant data included the author, year, and country, aims of the study, methods and research design, sample size, and the most common long COVID physical symptoms (Appendix A).

### 2.5. Collating, Summarising and Reporting 

Following data extraction and the formulation of the respective tables, including Appendix A, a narrative synthesis was undertaken to address the literature review research question. 

## 3. Results

### 3.1. Study Selection and Characteristics

From the database searches on long COVID, 7403 studies in total were identified (Figure 2). A total of 751 studies were available for full-text review following the removal of duplicates and exclusion of records based on title and abstract screening. A total of 679 studies were further excluded as they were not original research (review studies) and/or not peer reviewed. Further excluded studies were those where more than 25% of the participants were hospitalised, the patients’ symptoms were not physical or the data collection process was unclear, leaving a final selection of 70 studies meeting the inclusion criteria. Of the included studies, 60 were cohort studies; eight were case reports; and two were case series. The studies represented 289,199 patients in total. 

### 3.2. Description of Included Studies 

No exclusion dates were applied due to the emergent nature of the COVID-19 virus. The earliest included study was published in November 2020 by Townsend (2020), whilst the majority of the studies were published in 2021. The range of participants in the 60 cohort studies varied from 106,578 participants in a retrospective observational cohort study in the UK, using electronic health records for data collection, to smaller cohort studies, such as one conducted retrospectively in the US, with 49 participants. More than 50 of the 60 cohort studies had 100 participants or more (Appendix A). Many countries were included in the review: 33 studies were from Europe, 19 studies from the USA, nine studies from the Middle East, six studies from Southeast Asia, one study from Mexico, one study from Nigeria and one study from Australia (Appendix A).

### 3.3. Description of the Physical Symptoms of Long COVID from the Cohort Studies

Interpreting the results was difficult due to the wide range of reporting mechanisms. This review has remained true to the individual reporting mechanisms of each study, hence the apparent discrepancy between the numbers of studies included in the summary tables and text. 

Fatigue and shortness of breath were the main reported symptoms of long COVID (Figure 3). 92.6% of the 162,282 participants from 55 of the studies presented with fatigue, followed by 81.8% reporting shortness of breath (SOB) (Table 3). Thirty-nine of the 57 cohort studies reported that fatigue was the most common long COVID symptom. Eleven of the 57 studies reported that breathing difficulties (breathlessness, SOB) were the most common long COVID symptom, and two studies reported that muscle and joint pain were the most reported symptom of long COVID (Table 3). 

The mean age reported in 41 of the studies was 43.5 years (Table 4). Ten of the studies reported the median age as 47 years. The remaining six studies only alluded to age groups, but not specific ages, the age group most documented being 35–45 years. Forty-five out of the 57 studies indicated that over 50% of the long COVID suffers were female. Only 20 studies reported the participants’ smoking status and body mass index (BMI). However, within those studies, on average, 23.7% of the long COVID participants were active smokers and the average BMI was 27.13, i.e., in the “overweight” range [30] (Table 3).

**Figure 3 healthcare-10-02577-f003:**
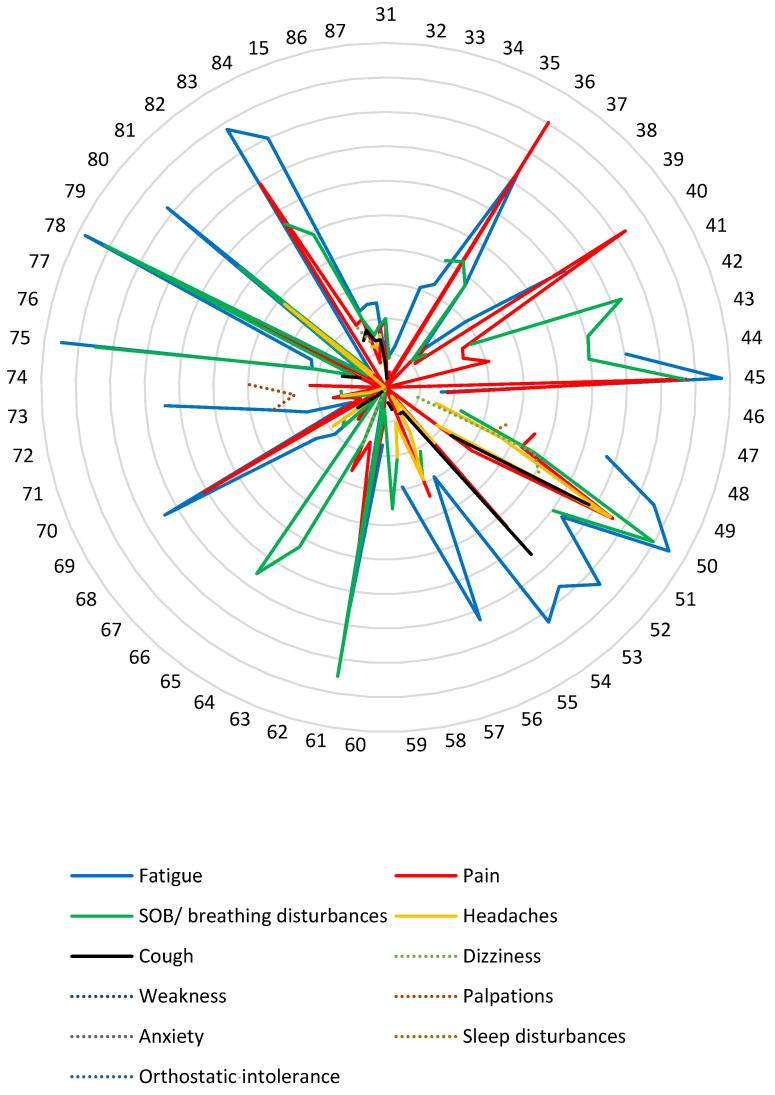
Radar chart illustrating the prevalence of the specific long COVID symptoms in the cohort studies with study number indicated on the circle circumference [15,31,32,33,34,35,36,37,38,39,40,41,42,43,44,45,46,47,48,49,50,51,52,53,54,55,56,57,58,59,60,61,62,63,64,65,66,67,68,69,70,71,72,73,74,75,76,77,78,79,80,81,82,83,84,85,86,87]. Appendix A illustrates how this radar chart was formulated.

A further three references were included in the scoping review, Appendix A. However, they do not appear on the radar chart due to the fact that they did not present numbers that allowed extrapolation into percentages [88,89,90]. However, they are discussed in other ways and included in the results.

The case reports described symptoms such as fatigue, SOB and joint pain of varying degrees. However, each case report described a different syndrome, such as multi-system inflammatory sequelae, Hashimoto thyroiditis, Graves’ disease, and subacute thyroiditis, Postural Orthostatic Tachycardia Syndrome (POTS), pulmonary fungal infection, Kawasaki-Like Syndrome, neuropathic pain and Anti-Neutrophil Cytoplasmic Autoantibody (ANCA) vasculitis. These syndromes indicate the long COVID symptoms that the individual participants suffered (Table 5).

## 4. Discussion

This scoping review describes the commonly reported physical symptoms of long COVID as reported in 72 primary research studies, including over 150,000 participants. The data from the cohort studies showed that the most common physical symptoms of long COVID in the non-hospitalised adult population were fatigue, shortness of breath (SOB), cough, muscle pain and joint pain. 

The physical symptoms reported in the cohort studies included neurological (fatigue, back pain, dizziness, sleep disturbances, headache, and weakness), respiratory (shortness of breath and cough), musculoskeletal (muscle and joint pain) and cardiac (orthostatic intolerance, palpations) systems. This reflects the specific impairments and limitations people with long COVID suffer from, thereby affecting their long-term health and quality of life. Identifying the specific physical symptoms will help health policy makers understand the specific services that are needed for this growing community. Thus, when considering the improvement of long COVID management services, many departments within the healthcare systems may require additional personnel and financial support to provide services that are more comprehensive.

The earliest article In this scoping review was from October 2020 [50]. This study assessed the recovery from symptoms following the onset of COVID-19 and was published prior to the WHO and NICE guidelines/definitions of long COVID. This article was published and freely accessible, a common practice during this period when the WHO and the global medical community encouraged scientific and medical researchers to provide and share information freely. At this time, the value of documenting the symptoms of long COVID was to assist medical professionals in identifying long COVID patients and help guide their ongoing treatment. However, this scoping review adds a further level of understanding. It contextualises the studies within a timeframe, using an accepted definition of long COVID, and highlights that to treat long COVID patients effectively, appropriate testing to rule out alternative diagnoses is essential. 

It is notable that the most commonly reported symptoms are found in the mid-forties age group. This possibly corresponds with some of the most highly experienced members of the workforce and the loss of this group from the workforce is likely to be extremely damaging across all industries and within the community. Furthermore, the average percentage of women participants in the cohort studies was 56.1% (S.D +/− 16.9), but care should be taken in interpreting this figure due to the way the studies were designed [35]. In the online surveys, gender was not an inclusion criterion. Current literature, however, suggests that women are more likely to respond to online surveys than men are; this trend has not been validated by rigorous research [101]. Further research is therefore required to clarify whether females are more likely to be affected by long COVID than males or whether these results are due to selection bias.

Within these studies, the average BMI was 27.13, falling within the overweight range (25.0–29.9) [102]. This led to the suggestion that overweight women in their mid-40s are the most likely candidates to have long COVID. The literature does not offer any specific reason as to why overweight people may be more susceptible to developing long COVID. However, these results do carry public health implications. Greater investment into health promotion programs that reinforce the benefits of reducing body fat (through healthy eating and regular exercise) could reduce the likelihood of long COVID following mild COVID-19. Identifying the most likely candidates for developing long COVID could aid health care professionals in giving a more accurate diagnosis and possibly lead to more appropriate treatment. Furthermore, health services can prioritise and support this population group in order to provide high quality care. 

In contrast to the cohort studies, the case reports/studies described long COVID patient syndromes. Some of the reports described the varied symptoms and attributed them to known syndromes, denoting the multi-organ dysfunction that acute COVID-19 caused even in mild non-hospitalised adult patients [103]. These published case studies reported long term respiratory complications [94] cardiac involvement [93,100], neurological dysfunction [96], and autoimmune dysfunction [97]. However, the symptoms of these syndromes did reflect the physical symptoms described in the cohort studies. For example, anti-neutrophil cytoplasmic autoantibody (ANCA) vasculitis, as described by Morris [97], in a 53-year-old patient who presented one month following his mild acute COVID-19 infection with symptoms including fatigue, generalised muscle pain, and loss of appetite. Another example is the postural tachycardia syndrome (POTS), syndrome described by Johansson [93], in a 42-year-old woman presenting three months following acute COVID-19 infection, with symptoms including fatigue, chest pain, muscle aches and postural light-headedness. The case studies did reinforce the gender distribution, as six of the eleven cases described were female. The cohort studies highlighted the variety of persistent physical symptoms observed in long COVID and the case studies supported these findings by describing symptoms that are more specific. The case studies therefore provided a better understanding of long COVID physical symptoms by catagorising them into syndromes.

A variable in both the cohort studies and the case studies was the number of days post-initial onset of the acute infection. This disparity reflects the differences in the criteria used in the studies to define long COVID. It also supports the marked variation of physical symptoms that were reported over different periods of time; the shortest being 30 days post-acute infection onset [97] and the longest being a median of 217 days post-acute infection onset [83]. The disparity of prevalence of the long COVD physical symptoms in this scoping review reflects the differences in the definitions of long COVID. As there was, and still is, uncertainty about the definition of long COVID, research inclusion and exclusion criteria for any post-COVID diagnosis differ between studies. It is important to consider that the current definition of long COVID, as defined by the NICE guidelines (2020) and WHO (2021), has two aspects. Firstly, persisting or new symptoms following acute COVID-19 and, secondly, the fact that the symptoms cannot be explained by another alterative diagnosis. Whilst this scoping review has identified the common physical symptoms associated with long COVID, there can be no certainty that the symptoms could not be explained by another diagnosis. Research on this topic is therefore fundamental in aiding healthcare workers to understand the implications of an accurate diagnosis of long COVID. Although someone may present with persisting or new symptoms following their acute COVID-19 infection, a thorough investigation into alternative diagnoses must be considered before confirming a diagnosis of long COVID. Post-acute infection fatigue syndromes have been described for other viruses and bacteria [104,105,106,107]. Ongoing low-grade inflammation has been suggested as a cause of these symptoms, but the pathology remains largely unknown, and treatments are based on symptom relief. 

This scoping review identifies the profile of the physical symptoms of long COVID in patients who experienced mild COVID-19 and highlights their apparent similarities to symptoms reported in other documented syndromes. Long COVID would appear to share similar post-infectious disease symptoms such as those seen following Severe Acute Respiratory Syndrome (SARS) [107,108], Q-fever, glandular fever, epidemic polyarthritis, and Legionnaires disease [104,105]. The overlapping symptoms of fatigue, pain and SOB are also present in Myalgic Encephalomyelitis/Chronic Fatigue Syndrome (ME/CFS) and POTS, which are known post-infectious disease syndromes [104,106,109]. Understanding the similarities between long COVID and ME/CFS and POTS symptoms may provide clues about the mechanism of long COVID, and thus help inform its management and recovery pathways. As an example, physiotherapy rehabilitation strategies used for ME/CFS, and POTS could be used to inform physiotherapy rehabilitation programs for long COVID patients [33,110,111].

As seen in Appendix A, the research groups were from a number of different countries; the majority being from Europe, followed by the USA, and smaller numbers from the Middle East, Southeast Asia, Mexico, Nigeria, and Australia. As studies not in English were excluded, this could be seen as a limitation of this study. However, the suggestion that governments used legislation to silence journalists and media in a number of countries [112] may have prevented studies being published from these countries. These political issues, important as they are, are beyond the scope of this literature review.

Authors should discuss the results and how they can be interpreted from the perspective of previous studies and of the working hypotheses. The findings and their implications should be discussed in the broadest context possible. Future research directions may also be highlighted.

## 5. Strengths and Limitations

The extent of the scoping review and the large number of studies retrieved increased the validity of the search, ensuring that the results are both reliable and meaningful to end users. Limitations of this study include the heterogeneity of the study designs, settings, and populations, follow-up time and methods of symptom identification. All the cohort studies were observational studies; however, 21 were retrospective and 25 were cross-sectional prospective studies. The data from the electronic health records (her) used for the retrospective studies were entered by clinicians, whilst the data gathered for the majority of the cross-sectional studies were provided directly by the patients. It must also be taken into account that clinicians may have used different terminology, and/or put emphasis on different types of symptoms and clinical signs when recording medical history. Both study types do, however, report similar symptoms. Additionally, the vast amount of data collected from EHRs can be challenging for the researcher to use in a “meaningful” way to answer specific research questions [113]. All but three of the retrospective studies combined EHRs with surveys, interviews, physical assessment, or a Delphi Consensus, ultimately adding to the depth of understanding and interpretation of the data from the EHRs (Appendix A). Some of the surveys included reliable and validated assessment tools, such as The Chalder Fatigue Scale (CFQ 11) [114], The RAND-36 Measure of Health-Related Quality of Life [115], and the EQ-5D-3L [116]. The combination of both quantitative and qualitative data drawn from these studies may lead to improved generalisability of the results. To gain a better understanding of this complex issue, a combination of many different approaches is required. 

The heterogeneity of the studies is demonstrated by two prospective observational studies, one with 100 participants [84] and the other with 106,578 participants [81]. Vanichkachorn and colleagues used medical record reviews and focused interview patient-reported histories to gain their data, whilst the Taquet group study used only EHRs. The data gathered by Vanichkachorn et al. was both qualitative and quantitative and gained a more in-depth insight into the experiences of patients regarding long COVID symptoms. Taquet et al.’s study was purely quantitative, but used a much larger number of patients, and also involved influenza patients as a comparison group of matched controls for the COVID-19 patients, all diagnosed during the same time-period. This was one of the few research projects that attempted to include a control group, the majority having no controls, attributing all symptoms solely to long COVID.

From the literature search, 268 retrieved studies discussed physical symptoms in hospitalised patients (Figure 2). A very limited number of studies discussed non-hospitalised patients alone, suggesting that long COVID may be a complication of the milder form of COVID-19 and, as such, an under-reported research topic. As mild COVID-19 is responsible for more than 80% of all confirmed cases of COVID-19, it is important that more attention be directed to this group of patients [18]. In order to avoid any biases with the hospitalised patient groups, the studies included in this review were required to have no more than 25% of hospitalised participants; for example, in the study by Danesh et al., (2021), 23% of the participants were hospitalised during the acute phase of their COVID-19 infection, possibly leading to “confirmation bias”. 

Additional limitations from the studies were the inconsistent terminologies used to describe the physical symptoms [81,87] and the definition of the severity of COVID-19 [82]. The fact that definitions of long COVID were published separately by NICE and WHO in November 2020 and September 2021, respectively, further hindered consensus and comparison between studies. Research produced before these dates or from researchers who did not adhere to the standardised definitions may explain the variability observed in the type and frequency of symptoms reported, thus limiting clinical interpretation [117]. This disparity is highlighted in the study by Davis et al. (2021), who surveyed their participants at a mean of 210 days post-acute infection, whilst another study assessed their participants at an average at 30 days post-acute infection [49]. Comparison between studies and interpretation of the aggregated data was rendered difficult due to these discrepancies. 

## 6. Conclusions

This scoping review focused on the impairment and limitations of “body structures” and “functions”, as defined by the International Classification of Functioning, Disability and Health framework (World Health Organisation, 2001), in individuals diagnosed with long COVID. Although a wide variety of physical symptoms were revealed in the review, the most commonly reported were fatigue, breathlessness/SOB, and pain. The similarities observed between long COVID and other post-acute infection syndromes may help in formulating ways to manage and promote recovery for long COVID. Inconsistencies were demonstrated in the research due to a possible lack of rigor, with researchers not always adhering to the standardised definitions of long COVID.

## Figures and Tables

**Figure 1 healthcare-10-02577-f001:**
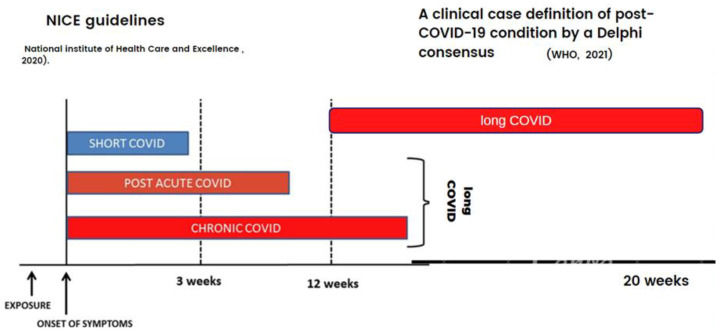
Classification of long COVID based on timing from symptom onset according to the NICE guidelines [11] and the WHO definition [9].

**Figure 2 healthcare-10-02577-f002:**
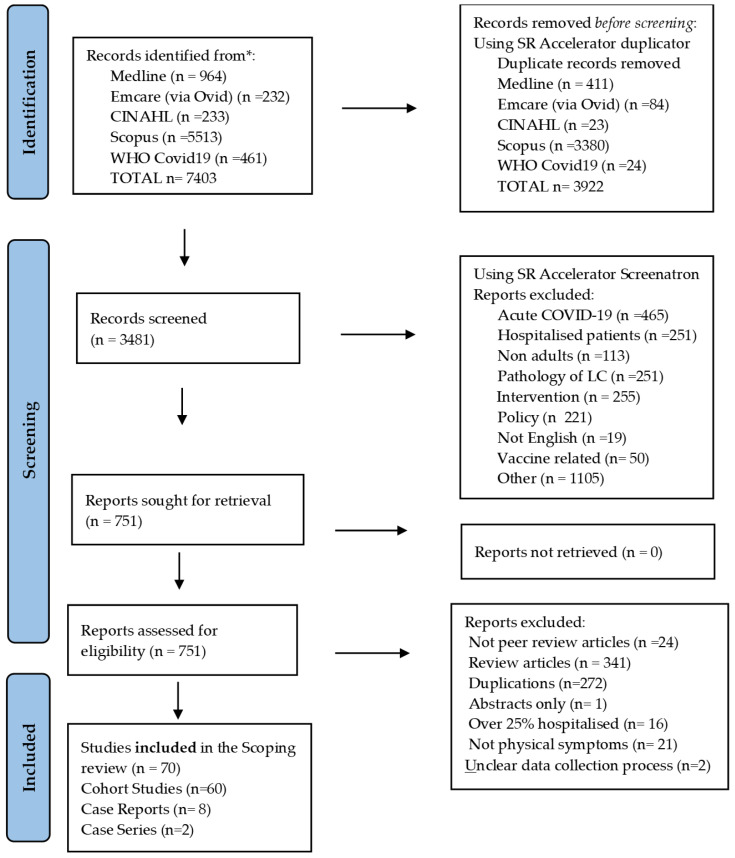
The PRISMA Flow Chart describing the search strategy. Page et al., 2020 [28].

**Table 1 healthcare-10-02577-t001:** 50 of the most common post-acute COVID-19 patient symptoms found in electronic health record clinical notes.

Pain	Insomnia
Anxiety	Pain in extremities
Depression	Paresthesia
Fatigue	Peripheral edema
Joint pain	Palpations
Shortness of breath	Diarrhea
Head aches	Itching
Nauseaand or vomiting	Erythema
Myalgia	Lower urinary tract symptoms
Gastroesophageal reflux	Lymphadenopathy
Cough	Edema
Back pain	Weight gain
stress	Sino-nasal congestion
Fever	Pain in throat
Swelling	Abnormal gait
Bleeding	Respiratory distress
Weight loss	Visual changes
Abdominal pain	Chills
Dizziness or vertigo	Urinary incontinence
Chest pain	Sleep apnea
Weakness	Confusion
Constipation	Hearing loss
Skin lesions	Problems with taste and smell
Wheezing	Difficulty in swallowing
Rash	Loss of appetite

Adapted from Wang et al., 2021 [15].

**Table 2 healthcare-10-02577-t002:** Inclusion and exclusion criteria for study selection.

Inclusion Criteria	Exclusion Criteria
Papers which discussed patients with long COVID (and all synonyms as described in the search term section)	
>75% of the participants in the study were non hospitalised	>25% of participants were hospitalised
>75% of the participant were between the ages of 18–65	Participants were <16 or >65 years
Primary research	Not primary research (i.e., reviews)
Physical signs and symptoms	Primarily reporting cognitive, psychological, or social signs and symptoms
Data could be clearly extracted	Data that could not be clearly extracted
Population was clearly defined	Population was not clearly defined
Papers written in English	Papers not written in English

**Table 3 healthcare-10-02577-t003:** Summary of percentages of long COVID symptoms as indicated in each study.

Symptom	Number of Studies Reporting Symptom/Total Number of Studies (%)	Number of Patients with Symptom/Total Number of Patients
Fatigue	51/55 (92.7%)	21,129/162,282
Shortness of Breath (SOB)	45/55 (81.8%)	23,109/162,282
Chest Pain	29/55 (52.7%)	10,463/162,282
Cough	27/55 (49%)	4678/162,282
Headaches	26/55 (42%)	17,991/162,282
Muscle Ache	24/55 (43.6%)	8284/162,282
Joint Pain	19/55 (34.5%)	6778/162,282
Sleep Disturbances	11/55 (20%)	6778/162,282
Dizziness	10/55 (18.1%)	4285/162,282
Palpation	9/55 (16.3%)	3261/162,282
Anxiety	8/55 (14.5%)	173/162,282
Muscle and joint pain	8/55 (14.5%)	7534/162,282
Weakness	2/55 (3.6%)	3236/162,282
Back pain	2/55 (3.6%)	4429/162,282
Orthostatic Intolerance	1/55 (1.18%)	16/162,282

**Table 4 healthcare-10-02577-t004:** Summary of demographics from the cohort studies [13,31,32,33,34,35,36,37,38,39,40,41,42,43,44,45,46,47,48,49,50,51,52,53,54,55,56,57,58,59,60,61,62,63,64,65,66,67,68,69,70,71,72,73,74,75,76,77,78,79,80,81,82,83,84,85,86,87].

Characteristics		Relevant Studies
Percentage of females Number of studies with over 75% women	64.48% 8 studies	[31,32,33,34,35,36,37,38,39,40,41,42,43,44,45,46,47,48,49,50,51,52,54,55,56,57,58,59,60,61,62,63,64,65,66,67,68,69,70,71,72,73,74,75,76,77,78,79,80,81,82,83,84,85] [35,38,44,50,52,69,78,83]
Mean age	43.5 years (mean)	[15,34,35,38,40,41,42,45,47,48,49,50,51,52,54,55,56,57,61,62,64,65,67,70,71,72,73,74,75,76,77,78,79,80,81,82,83,84]
Median age	47 years (median)	[31,32,33,36,39,58,59,60,66]
Mean % of population were smokers with long COVID	23.7%	[32,35,37,38,40,45,56,60,61,64,65,68,70,70,71,75,77,82]

**Table 5 healthcare-10-02577-t005:** Summary table of the Case Reports and Case Series.

Author/Country/Year	Condition Described as Part of the Long COVID Syndrome	Sample Size	Study Design
Balan, et al., USA 2021 [91]	Multisystem inflammatory sequelae.	1 male	Case report
Feghali, USA 2021 [92]	Hashimoto thyroiditis, Graves’ disease, and subacute thyroiditis	3 females	Case series
Johansson, Sweden, 2021 [93]	POTS—Postural Orthostatic Tachcardia Syndrome	2 females, 1 male	Case series
Kakamad, Iraq, 2021 [94]	Pulmonary fungal infection	1 male	Case report
Lechien, France, 2021 [95]	Kawasaki-Like Syndrome	1 male	Case report
McWilliam UK, 2021 [96]	Neuropathic pain	1 male	Case report
Morris, D., USA, 2021 [97]	ANCA vasculitis	1 male	Case report
Omololu, Nigeria, 2021 [98]	Persistent symptoms post COVID-19	1 male	Case report
Taribagil., et al., UK, 2021 [99]	Presentation of a patient with long COVID	1 female	Case report
Vera-Lastra, et al., Mexico, 2021 [100]	Fluctuating symptoms of myopericarditis	1 male	Case Report

## Data Availability

The supplementary tables containing the data supporting these results have been attached and will be made available online.

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
