# Peer review of "Most Common Long COVID Physical Symptoms in Working Age Adults Who Experienced Mild COVID-19 Infection: A Scoping Review"

_healthcare, 2022, doi:10.3390/healthcare10122577_

Round 1
Reviewer 1 Report
In this scoping review, the authors investigated the symptoms of long-COVID in adults with mild infection. The results may provide reference for the treatment and management of long-COVID which is a public health need. Overall, it is clearly structured. However, some improvements are essential to meet publishable standards.
1. There are too many colors in Figure 3 where the incidences of symptoms are highly different. I recommend the categorize these symptoms according to their highest reported incidence, and draw 2 or 3 subfigures instead of listing all in one figure.
2. Table 4 is confusion. It seems the title is problematic. Please update.
3. There is a lack of in-depth discussion on the underlying causes on these symptoms. Some existing studies have attempted explainable classification of COVID-19 severity (Refer: 10.3390/ijerph191710665), while the comprehensive analysis in long-COVID cohorts is still a research gap. The results of this review provide reference to enable further pathophysiological investigation.
Author Response
Thank you for your comments and recommendations. I have addressed all the comments and attached a table that clearly presents the comments and the amendments.
Please see the attachment
Kind regards
Zoe Mass Kokolevich

Reviewer 2 Report
It's an honour to review the article "healthcare-2033250 ", titled “Commonly reported physical symptoms of long COVID in adults aged 18 – 65 years who experienced a mild COVID-19 infection: a scoping review” for “Healthcare” journal.
The manuscript is original, interesting, and sometimes pleasant to read. It’s suitable for publication. However, there are some improvements that authors should do in order to improve the overall level of their article to merit publication in this valuable journal. Here are my comments:
1. Please rephrase from line 12 to 15 as it doesn't read well. Please also consider including a more general statement.
2. Line 18, COVID-19 instead of Covid-19
3. Line 31, please write something like "severe acute respiratory syndrome coronavirus (SARS-CoV-2)" causes the Coronavirus Pandemic (COVID-19)" as it is the first it appears in the text, you need to write it in full before using the dimunitive.
4. Line 32-34 you need to include more studies for your general statement, I think you have also to add at least 1 or 2 sentences about the situation in the world writing about preventive measures, socio-economic effects...etc. Here are some papers to cite in your paper:
*https://doi.org/10.1038/s41591-022-01750-1
*https://doi.org/10.3390/healthcare10071341
*https://doi.org/10.1038/s41579-021-00639-z
*https://doi.org/10.1016/j.bbi.2021.12.020
*https://doi.org/10.3390/ijerph19159586
*https://doi.org/10.1136/bmj-2021-069676
5. Line 43, NICE guidance ? please write The National Institute for Health and Care Excellence (NICE)
6. Line 71, please refer to authors' instructions, for example Wang et al. [8] as an example.
7. Line 108, same as the last comment, please apply it in the whole document
8. Concerning your search strategy, did you followed an existing protocol or used your own one ?
9. Why not use Web of Science database ?
10. Table 4, why did you put a line after the first symptom ?
11. Figure 3, why 26 is not in the same line ?
12. Line 331, please remove extraparentheses
13. Please apply authors instructions in the website of the journal, as an example, references should be [XX] instead of (XX) for example.
14. My major comment is related to English language, there are a lot of run-on sentences and grammatical mistakes, please try to make a revision by a native English, a specialist or an English editing service.
Best wishes.
Author Response
Thank you for your comments and recommendations. I have addressed all the comments and attached a table that clearly presents the comments and the amendments.
Please see the attachment.
Kind Regards
Zoe Mass Kokolevich

Round 2
Reviewer 1 Report
Thanks for the update. Some of my earlier comments have been well addressed while others need your further work. Besides, many details deserve improvement.
1. Regarding figure 3, first, please delete the gray boarders. Second, it is still unclear due to the overlapping of colors. There is a heterogeneity among studies in percentage. I suggest to draw a box plot where the symptoms are listed according to the mean/median of frequency (from high to low).
2. The format of table 4 is problematic. Please double check. Make a three-line table.
3. Table 1 only lists the name of symptoms. Please add another column about their definition/diagnostic criteria. If possible, use another column to list the relevant studies.
4. Figure 2 is difficult to read. Please enlarge the font.
Author Response
Dear Reviewer number 1,
I am sharing with you the table with our response to the further requests that you suggested and we have had the manuscript edited by a professional English language editor.
We hope the manuscript now satisfies the reviewers' comments.
I have attached the revised manuscript version with all the amendments and the table.
Kind regards
Zoe Mass Kokolevich
